# Clinical Efficacy of Three-Dimensional-Printed Pure Titanium Fracture Plates with Locking Screw Systems in Distal Tibia Fractures

**DOI:** 10.3390/medicina61010137

**Published:** 2025-01-15

**Authors:** Ji Hye Choi, Jun Hyoung Lee, Seung Hyeop Lee, Woo Young Jang

**Affiliations:** 1Department of Orthopedic Surgery, Anam Hospital, Korea University College of Medicine, 73 Goryeodae-ro Seongbuk-gu, Seoul 02841, Republic of Korea; iammeddukk@gmail.com; 2Department of Orthopedic Surgery, Korea University College of Medicine, Seoul 02841, Republic of Korea; jhlee2@implantcastasia.com; 3Cubelabs, Inc., Seoul 02632, Republic of Korea; shlee@cubelabs.co.kr; 4Institute of Nano, Regeneration, Reconstruction, Korea University, 73 Goryeodae-ro, Seongbuk-gu, Seoul 02841, Republic of Korea

**Keywords:** titanium alloy, pure titanium, 3D printing, fracture, orthopedic implant

## Abstract

*Background and Objectives:* Distal tibia fractures are high-energy injuries characterized by a mismatch between standard plate designs and the patient’s specific anatomical bone structure, which can lead to severe soft tissue damage. Recent advancements have focused on the development of customized metal plates using three-dimensional (3D) printing technology. However, 3D-printed metal plates using titanium alloys have not incorporated a locking system due to the brittleness of these alloys. Therefore, this study aimed to determine whether a locking mechanism can be effectively implemented using 3D-printed pure titanium and further evaluate the clinical outcomes of such implants in patients with distal tibia fractures. *Materials and Methods*: Between March 2021 and June 2022, nine patients who underwent open reduction and internal fixation for distal tibia fractures using 3D-printed pure titanium plates were enrolled. Pure titanium powder (Ti Gr.2, Type A, 3D Systems, USA) was spread to a thickness of 30 μm and partially sintered using a 500 W laser to produce the 3D-printed metal plates. The locking screws were fabricated using a milling process. Open reduction and internal fixation were performed on the nine patients using 10 customized plates. The clinical efficacy was analyzed using the union rate, and complications, such as infection and skin irritation, were evaluated to ensure a comprehensive outcome assessment. *Results:* Surgical treatment was successfully performed on nine patients, with nine of ten plates remaining stable and undamaged. However, one patient with neurofibromatosis experienced a fractured metal plate, which necessitated revision surgery using a metal rod. No screw loosening or surgical wound complications occurred. *Conclusions:* This study showed that 3D-printed pure titanium plates with integrated locking screw systems provide a viable and effective solution for managing distal tibia fractures. Three-dimensional printing and pure titanium show promise for orthopedic advancements.

## 1. Introduction

Distal tibia fractures are high-energy injuries characterized by the presence of multiple fracture fragments, often resulting from significant traumatic events. These complex fractures present considerable challenges in terms of management due to their intricate nature and the substantial forces involved [1,2,3]. Their treatment is further complicated by the need for surgical fixation, particularly when utilizing metal plates on the medial aspect of the tibia [4]. A critical issue arises from the mismatch between standard plate designs and the patient’s specific anatomical bone structure, which can lead to severe soft tissue damage. Such discrepancies increase the risk of complications, including nonunion, osteomyelitis, and other significant adverse outcomes, thereby affecting patient recovery considerably.

In response to these challenges, recent advancements have focused on the development of customized metal plates using three-dimensional (3D) printing technology. This approach aims to achieve a superior fit to the patient’s unique bone anatomy, thereby enhancing implant stability and minimizing complications [5,6,7]. Although titanium alloys have been the traditional material of choice in orthopedic surgery due to their favorable properties, their inherent brittleness poses substantial limitations. Specifically, the milling process required to implement locking mechanisms often induces cracking in titanium alloys, leading to repeated failures and inadequate fixation.

Pure titanium, by contrast, offers a distinct advantage owing to its lower stiffness and higher ductility, which may facilitate the successful integration of locking screw systems through 3D printing techniques. The enhanced elongation characteristics of pure titanium could lead to improved load transfer between the implant and the surrounding bone, thereby promoting long-term stability and osseointegration [6]. Despite these potential benefits, the use of pure titanium has been restricted by concerns regarding its mechanical strength and the associated risk of fracture under physiological loads [8]. Because of this concern, the development of repeated heat treatments improved the mechanical properties of pure titanium, which our study introduced to improve the strength and physical properties of pure titanium plates [9].

To date, there has been a report documenting the use of 3D-printed pure titanium plates [10]; however, no reports have documented the use of 3D-printed pure titanium plates specifically designed for the management of distal tibial fractures. Also, there have been attempts to compare locking thread strength in 3D printed locking threads, but they are far from locking threads for screw heads, and the studies lack clinical outcomes [11]. Therefore, this study aimed to determine whether a locking mechanism can be effectively implemented using 3D-printed pure titanium and to evaluate the clinical outcomes of such implants in patients with distal tibia fractures.

## 2. Materials and Methods

### 2.1. Participants

This prospective study (IRB No. 2021AN0053) was approved by the institutional review board of the tertiary referral medical center, and informed consent was obtained from all patients. Between March 2021 and June 2022, nine patients who underwent open reduction and internal fixation for distal tibia fractures using 3D-printed pure titanium plates were enrolled in this prospective study. The inclusion criteria were patients with distal tibia fractures who received open reduction and internal fixation with a 3D-printed custom tibia plate and had a postoperative follow-up within at least 12 months. The exclusion criteria were patients with soft tissues unsatisfactory for direct surgical exposure, infection, tumors, or any other condition in which the risks of surgery exceeded the expected benefits due to the patient’s general condition.

### 2.2. Designing the Fracture Plate

This process began by converting computed tomography (CT) data into 3D images of the patient’s bones using Mimics software (Version 24.0, Materialise, Leuven, Belgium). The CT (INGENUITY CT, Philips, Cleveland, OH, USA) images used in this study were acquired with a slice thickness of 1 mm to ensure precise contact between the patient’s bone and the fracture plate. Models of the patient’s CT data were utilized to establish a threshold range of 200 to 350 HU, enabling the extraction of valid data corresponding to the affected bone region. Subsequently, a 3D reconstruction process was employed to convert the extracted data into a 3D model suitable for design applications. To address surface irregularities arising from CT scan slice segmentation during the 3D reconstruction process, the Smooth Factor function in Mimics software (Version 24.0, Materialise, Leuven, Belgium) was applied at a fixed value of 0.6. Subsequently, 3-Matics software (Version 16.0, Materialise, Leuven, Belgium) was utilized to rearrange the fractured bones into the correct alignment, creating a suitable shape for each patient. The simulation of the optimal screw insertion position was conducted by utilizing three-dimensional morphology and medical imaging. The screw insertion pathway was designed by simulating the type (cortical or locking screw) and appropriate length of screws for fixation, as well as determining the angle of screw fixation to be perpendicular to the fracture site as possible to the bone, ensuring avoidance of the adjacent neurovascular structure, as well as stable fixation of the fracture plate within the body.

### 2.3. Fabrication of Pure Titanium Fracture Plate

The customized fracture plate was produced using the SLM-type metal 3D printer DMP Flex 350 (3D Systems, Rock Hill, SC, USA). Argon gas was introduced into the manufacturing chamber to maintain an oxygen concentration of 25 ppm, preventing oxygen inflow during the additive manufacturing process. The fabrication was conducted in a vacuum environment. Pure titanium powder (Ti Gr.2, Type A, 3D Systems, Rock Hill, SC, USA), free of aluminum and vanadium, was spread to a thickness of 30 μm and partially sintered with a 500 W laser. The bone plates were manufactured using a microstructure-controlled additive manufacturing method licensed from the Korea Institute of Industrial Technology. This technology optimizes laser scan speed (250–500 mm/s), spot distance (40–100 μm), and exposure time (80–400 μs) to induce isotropic α-phase microstructures in pure titanium. Additionally, layer thickness (20–100 μm) and particle size (10–100 μm) were controlled to improve grain size and orientation. Energy density adjustments were made to enhance microstructure homogeneity and reduce residual stress. Experimental validation using electron backscatter diffraction and Vickers hardness testing confirmed the relationship between the optimized parameters and the mechanical properties of the material [9,12].

Two post-treatments were performed. First, heat was supplied (KF-420, © K&F TECH, Ansan-si, Gyeongi-do, Republic of Korea) at 590 °C for approximately 2 h in a heat treatment furnace to promote the precipitation hardening of CP-Ti in the fine intermetallic compound structure, thereby enhancing the mechanical properties. Second, the locking thread was processed through computer numerical control (CNC) precision machining to increase fixation force, preventing screws from being pulled out or loosening.

All manufacturing procedures were performed in a manufacturing facility with a Good Manufacturing Practice (GMP) certificate certifying that the facility has passed the strict manufacturing process review of the Ministry of Food and Drug Safety, Korea’s medical device regulatory agency. High-pressure steam sterilization was confirmed before the operation.

### 2.4. Biomechanical Testing

To evaluate the strength of the plates, a static 4-point bending test and a dynamic 4-point fatigue test were performed according to the ASTM F382-17 standard [13]. Testing was conducted using a 10 kN fatigue tester (manufactured by Instron) on six samples of the manufactured plates with a center spacing of 19.5 mm and support spacing of 59.5 mm. In the static test, a bending force was applied at a rate of 5 mm/min at room temperature. For the dynamic 4-point fatigue test, 1,000,000 cycles were performed at 25% of the bending load at a frequency of 5 Hz at room temperature.

### 2.5. Clinical Efficacy Assessment

Clinical efficacy was analyzed using the union rate and the American Orthopedic Foot and Ankle Society (AOFAS) score. Additionally, complications, such as infection and skin irritation, were evaluated to ensure a comprehensive assessment of the treatment outcomes.

## 3. Results

### 3.1. Demographic Data

A total of nine patients were enrolled in this study, one of whom had bilateral distal tibia fractures, resulting in a total of 10 tibiae undergoing surgical reduction. Among these, three had closed fractures, and seven had open fractures. Three patients had hypertension, with one also having diabetes mellitus. One patient presented with neurofibromatosis type 1 and multiple clinical café au lait spots. No other severe underlying illnesses were observed (Table 1).

### 3.2. Pure Titanium Fracture Plate

The custom-made fracture plate was designed to match the curvature of the patient’s bone, utilizing between 8 and 15 fixing screws for secure fixation. Cortical screws with diameters of 3.0 and locking screws with diameters of 2.0 were employed. To ensure stable support for the fixing screws, the thickness of the fracture plate was designed to be 3.5 mm, with an approximate length of 170 cm (±2 cm) and a weight of approximately 25 g (±2 g) (Figure 1). The fracture plate was produced with a metal 3D printer using pure titanium as the raw material. During heat treatment aimed at compensating for the low strength of pure titanium, no significant changes in appearance or thermal shrinkage were observed. A hole processing jig was utilized to enhance the fixation force for the locking screw thread; subsequently, the desired locking screw thread was successfully achieved (Figure 2).

### 3.3. Biomechanical Testing

Biomechanical testing was conducted in accordance with ASTM F382-17 standards for five manufactured plates (Figure 3). The results of the four-point bending test indicated that the bending strength exceeded 750 N/mm^2^. Additionally, it was confirmed that the structural stiffness was maintained without deformation at levels greater than 200,000 N/mm^2^ (Table 2). In the dynamic four-point bending test, no deformation or damage was observed after applying a load equivalent to 25% of the bending strength over 1,000,000 cycles at a frequency of 5 Hz (Table 3).

### 3.4. Clinical Efficacy Assessment

Ten tibial fractures that underwent surgical reduction were followed up for 1–2 years. Over the follow-up period, in nine out of 10 tibiae, the fracture plate remained undamaged, deformed, and securely fixed within the body. The mean American Orthopedic Foot and Ankle Society (AOFAS) score was 81.3 (SD 5.7). In one patient, who experienced failed fracture healing and had neurofibromatosis type 1, the fracture plate did not fulfill its intended role in the case of an open fracture. Consequently, revision surgery was performed using a tibial intramedullary nail and additional bone grafting. A biopsy of the previously failed fracture site revealed chronic inflammation with fibrosis and dead bone particles (Figure 4).

## 4. Discussion

This study is among the first to explore the clinical applicability of pure titanium implants utilizing 3D printing technology. The results indicated that, out of 10 tibiae, nine had fracture plates that remained undamaged and were stably fixed. The findings from this study provide important insights into the clinical feasibility of utilizing 3D-printed pure titanium plates for distal tibia fractures. One of the primary advantages of employing 3D-printed pure titanium is its capacity to address anatomical mismatches between standard metal plates and patient-specific bone structures. This capability is particularly critical for distal tibia fractures, where the anatomical constraints of the medial tibia complicate proper fixation with conventional plates (Figure 5).

Locking screw systems play an essential role in enhancing implant stability, especially in the context of complex fractures and compromised bone quality. Using locking screws eliminates the friction that occurs on the periosteum after fracture plate fixation, allowing stable support within the body [14,15,16]. The integration of locking mechanisms into pure titanium plates was successfully achieved through 3D printing, suggesting that pure titanium’s lower stiffness and greater ductility can mitigate the challenges associated with brittle titanium alloys. This enables improved load distribution, reduced stress concentration at the screw–bone interface, and enhanced bone healing. The absence of screw loosening in all implants in this study further indicates the mechanical reliability of 3D-printed pure titanium for orthopedic applications.

A notable finding of this study is the potential of 3D-printed plates as personalized implants, which represents a significant advancement in orthopedic surgery beyond the context of distal tibia fractures. Using 3D printing technology to create patient-specific implants allows for a more individualized approach to fracture management. By ensuring a better fit between the implant and the patient’s anatomy, personalized implants can improve functional outcomes, shorten recovery times, and reduce the need for revision surgeries. The ability to design and customize 3D-printed fracture plates for each patient ensures optimal anatomical congruence, thereby enhancing stability and reducing discomfort. A study involving 673 patients demonstrated that the use of 3D printing in fracture management significantly improved perioperative outcomes compared to conventional open reduction and internal fixation techniques [6]. Patient-specific fracture plates have no economic advantage compared to conventional fracture plates; however, they reduce the number of reoperations and shorten recovery time, with several studies demonstrating potential benefits [17].

High-energy lower extremity trauma frequently results in soft tissue damage, which remains one of the most challenging issues for orthopedic surgeons. In this study, although seven patients had high-energy open fractures, no complications were related to soft tissue damage. The AOFAS score of 81.3 observed in this study reflects a good functional outcome, although it does not fall within the excellent range of 90 or above. This is consistent with the challenges of treating distal tibial fractures, particularly open fractures. All patients included in this study sustained high-energy trauma, such as traffic accidents, with 7 out of 10 cases involving open distal tibial fractures. A previous study has reported that open fractures are associated with severe vascular and soft tissue damage, leading to a higher incidence of complications such as wound dehiscence and soft tissue infections [4]. These factors fundamentally impair the soft tissue environment of the distal tibia, inherently limiting the potential for functional recovery and adversely affecting long-term outcomes. However, one procedure failure occurred in a patient with neurofibromatosis, indicating that the manufactured plate lacked sufficient strength to manage this condition (Figure 4). Future studies should combine pure titanium with other biocompatible materials or incorporate advanced surface treatments to enhance osseointegration further and assess its mechanical properties.

This study has several limitations that must be acknowledged. It was conducted at a single institution and involved a relatively short-term follow-up period, and the small sample size limits the statistical generalizability of its findings. Additionally, all surgeries were performed by one experienced orthopedic surgeon, which may introduce a bias in the results. These limitations underscore the need for larger multicenter studies involving diverse patient populations to validate the findings and expand their applicability. Despite these limitations, the main strength of this study lies in the successful creation of personalized fracture plates through 3D printing and the effective implementation of a locking screw system using pure titanium. This study demonstrates a novel approach by overcoming the limitations of titanium alloys in 3D printing and the mechanical weaknesses of pure titanium plates. By implementing precise screw threads and strengthening the material through optimized manufacturing processes, we have achieved sufficient mechanical properties for clinical application. Furthermore, the versatility of this approach enables its application to various plate geometries, underscoring its potential for broader clinical utility. This approach overcame the limitations of traditional fracture plates and enhanced stability in fracture treatment. The favorable mechanical properties of pure titanium, coupled with the precision of 3D printing, provide a promising direction for the development of advanced orthopedic implants.

## 5. Conclusions

In conclusion, this study revealed that 3D-printed pure titanium plates with integrated locking screw systems provide a viable and effective solution for managing distal tibia fractures. The customization afforded by 3D printing, coupled with the favorable properties of pure titanium, may hold considerable promise for advancing orthopedic care. Further research and development in this field are essential to fully realize the potential of 3D-printed implants and expand their therapeutic applications in orthopedics.

## Figures and Tables

**Figure 1 medicina-61-00137-f001:**
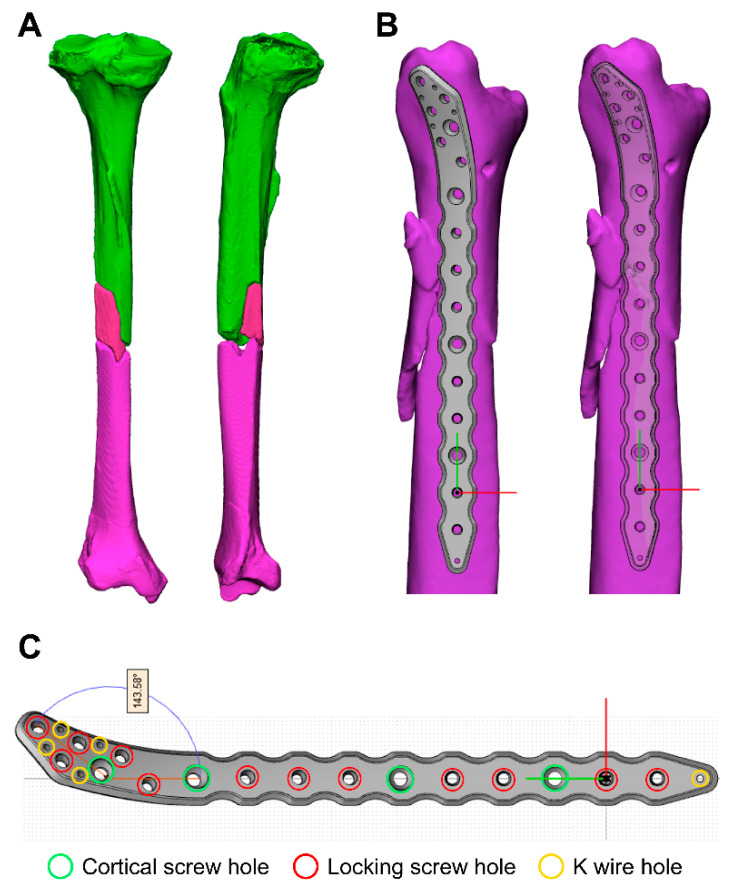
(**A**) Simulation model of the fractured tibia aligned correctly; (**B**) Custom fracture plate design upon simulation model; (**C**) Individual fixation plates designed with positioning screw holes for cortical and locking screws.

**Figure 2 medicina-61-00137-f002:**
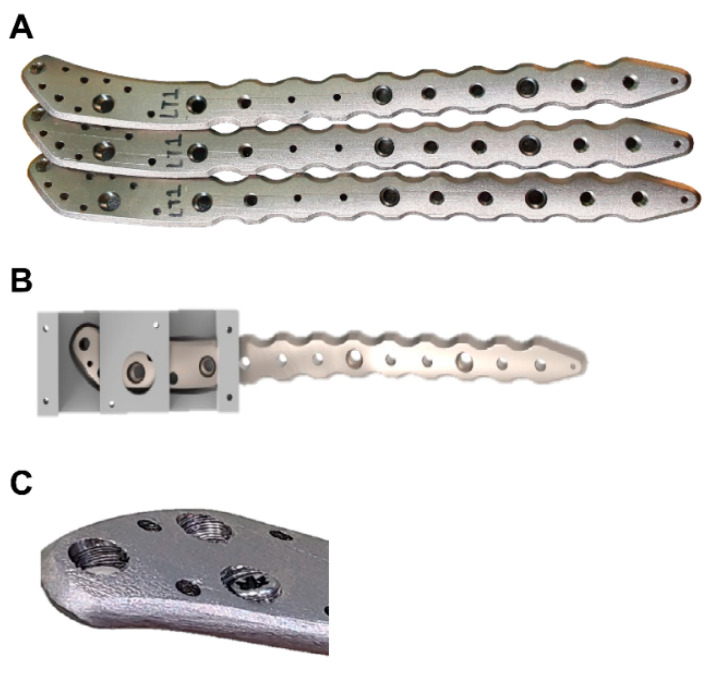
(**A**) Fracture plate fabricated through metal 3D printing; (**B**) Jig for locking screw thread; (**C**) Locking screw hole.

**Figure 3 medicina-61-00137-f003:**
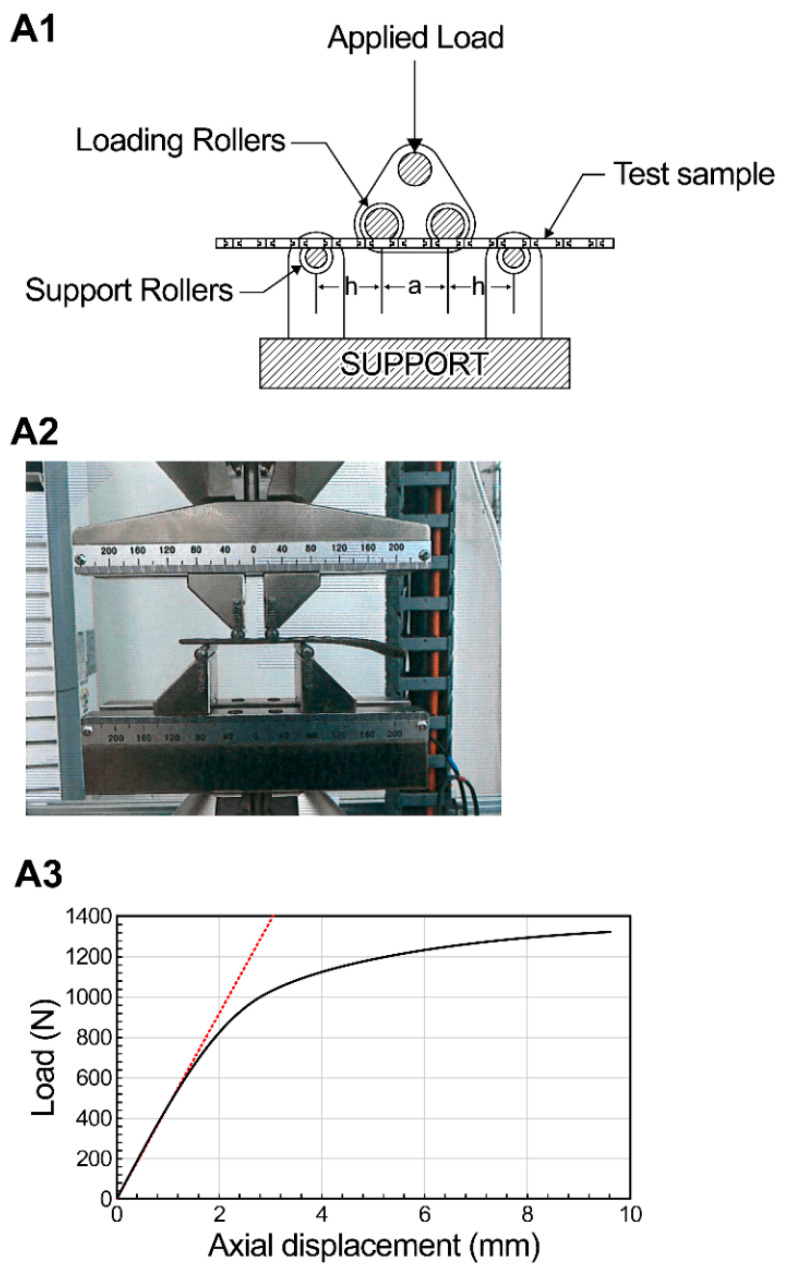
(**A1**) Test method according to ASTM F382-17DP; (**A2**) Fracture plate under actual test according to ASTM F382-17 test standard; (**A3**) Test results.

**Figure 4 medicina-61-00137-f004:**
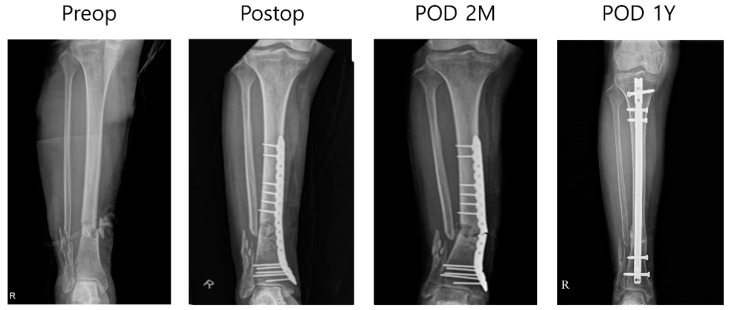
Failed fracture plate case: A 59-year-old female pedestrian was diagnosed with an open tibiofibular shaft fracture and subdural hematoma (SDH) following a traffic accident in which she was hit by a taxi. The patient had a medical history of neurofibromatosis. After massive irrigation and confirmed wound stabilization, minimally invasive percutaneous plating (MIPO) using a 3D-printed plate was performed 4 days after the trauma. The wound healed well without an additional coverage operation. Partial weight bearing was initiated 6 weeks post-operation. Three months post-operation, the patient presented to the outpatient clinic with lower leg pain; a plain radiograph confirmed plate breakage. She subsequently underwent reoperation with plate removal and conversion to an intramedullary (IM) nail fixation. An intraoperative biopsy of the nonunion site revealed findings consistent with chronic inflammation with fibrosis and the presence of dead bone fragments. After tibia IM nail fixation, fracture union was achieved, and the patient’s AOFAS score was 70. R means right sided.

**Figure 5 medicina-61-00137-f005:**
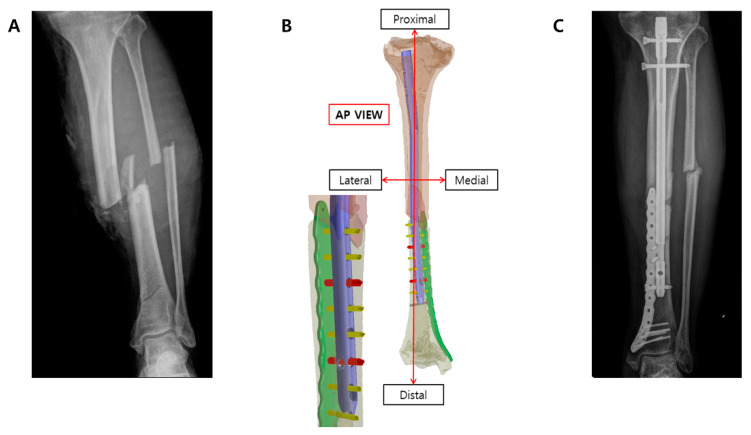
Customized fracture plate designed to be combined using a tibia intramedullary nail. (**A**) A 68-year-old male presented to the emergency department after sustaining an injury to his left lower leg with a heavy load. On arrival, the patient was diagnosed with an open tibiofibular shaft fracture. The patient had a medical history of hypertension. An additional fracture line was identified in the distal tibia, raising concerns that adequate stability could not be achieved with an intramedullary nail alone. (**B**) Therefore, supplementary plating was considered. (**C**) After massive irrigation and confirmed wound stabilization, closed reduction and internal fixation with a tibia intramedullary (IM) nail and additional 3D-printed plate fixation were performed 1 week after the trauma. The soft tissue healed well without an additional coverage operation. Partial weight bearing was initiated 6 weeks post-operation. A successful union was achieved with an AOFAS score of 70.

**Table 1 medicina-61-00137-t001:** Demographic data summary.

No. of patients	9
Age (years)	55.25 (SD 14.4)
Male: Female	7:2
BMI (kg/m^2^)	24.28 (SD 0.69)
Side (right:left)	6:4
Fracture characteristics	
Single tibia fracture/Multiple fracture	3:7
Closed fracture/Open fracture	3:7
Underlying medical history	
HTN	3
DM	1
Others (neurofibromatosis)	1
Values are presented as n or mean (SD)	

**Table 2 medicina-61-00137-t002:** Static four-point bending test results.

Specimen No.	Yield Load (N)	Bending Strength ^a^ (N∙mm)	Bending Stiffness (N∙mm)	Bending Structural Stiffness ^b^ (N∙mm)
#1	641.961	8345.493	467.30	3,264,246.267
#2	643.498	8365.474	466.37	3,257,749.907
#3	631.654	8211.502	469.96	3,282,827.253
#4	655.841	8525.933	466.91	3,261,521.987
#5	648.692	8432.996	467.02	3,262,290.373
Mean	644.921	8383.976	467.57	3,266,097.380
Std. Dev.	10.190	132.469	1.62	11,328.379

^a^ Bending strength is calculated using Formula A1.3 from ASTM F382-17. ^b^ Bending structural stiffness is calculated using Formula A1.1 from ASTM F382-17.

**Table 3 medicina-61-00137-t003:** Fatigue four-point bending test results.

Specimen No.	Maximum Load (N)	Minimum Load (N)	Cycles Completed	Remark
#7	162.0	16.2	1,000,000	No failure
#8	162.0	16.2	1,000,000	No failure
#9	162.0	16.2	1,000,000	No failure

## Data Availability

All data are fully available without restriction.

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
