# Peer review of "Clinical Efficacy of Three-Dimensional-Printed Pure Titanium Fracture Plates with Locking Screw Systems in Distal Tibia Fractures"

_medicina, 2025, doi:10.3390/medicina61010137_

Round 1

Reviewer 1 Report

Comments and Suggestions for Authors

Review on the article Clinical Efficacy of 3D Printed Pure Titanium Fracture Plates with Locking Screw Systems in Distal Tibia Fractures, submitted to Medicina.

The authors have attempted to evaluate the efficiency of 3D printed pure titanium fracture plates. While the study generally seems to be conducted properly, the manuscript does not provide necessary details that allows to reproduce the research or to evaluate the quality of prepared models. Moreover, small number of literature references does not provide appropriate background for the study. The authors must refer to these as well as additional comments presented below.

Introduction

Only 6 references were used as a background for the study, which is not sufficient. Moreover, only 2 out of these references were published within last 2 years.

Materials and methods

The description of any exclusion criteria is missing.

Information about CT images is missing (the number of images, distance between them, resolution etc.).

The description of modelling in Mimics software is missing (initial threshold, smoothing etc.).

A Figure presenting all bones modelled as well as individual fixation plates should be added to the manuscript.

While the authors claim “The optimal design was derived during this process by simulating the type and size of screws for fixation (…)”, there are no details regarding how optimal design was selected.

“known for its excellent precision among metal 3D printing products” – this description is redundant.

Results

It is unclear what authors mean by the “size” of the screw. Is it Screw diameter?

General dimensions of the fixation plates should be presented in any of included Figures.

Figures 3a and 3b presenting biomechanical testing setup should be presented in materials and methods section.

“the fracture plate remained undamaged, deformed” – did the authors mean that plates were undamaged and undeformed over time?

Author Response

Dear Editor:

We thank reviewers for thoughtful and constructive comments on our paper. Authors deeply appreciate and revised our manuscript considering the comments by the reviewers.

Reviewer #1:

The authors have attempted to evaluate the efficiency of 3D printed pure titanium fracture plates. While the study generally seems to be conducted properly, the manuscript does not provide necessary details that allows to reproduce the research or to evaluate the quality of prepared models. Moreover, small number of literature references does not provide appropriate background for the study. The authors must refer to these as well as additional comments presented below.

Introduction

Only 6 references were used as a background for the study, which is not sufficient. Moreover, only 2 out of these references were published within last 2 years.

We appreciate your comment. We noticed that a few of the references are outdated. References before year 2010 have been revised based on recently published papers.

Materials and methods

The description of any exclusion criteria is missing.

Thank you for your thorough review of this paper. We have revised the Participants section of the Materials and methods. Added revised section starts as below:

Exclusion criteria were patients with soft tissues unsatisfactory for direct surgical exposure, infection, tumor, or any other condition that risks of surgery exceed expected benefits due to patients’ general condition.

Information about CT images is missing (the number of images, distance between them, resolution etc.).

The description of modelling in Mimics software is missing (initial threshold, smoothing etc.).

We appreciate your comment. We have included additional information of CT images used in this study along with additional description of modelling in Mimics software. Added “Designing the Fracture Plate” of the materials and method section starts as below:

This process began by converting computed tomography (CT) data into 3D images of the patients’ bones using Mimics software (Materialise, Leuven, Belgium). The CT (INGENUITY CT, Philips,) images used in this study were acquired with a slice thickness of 1mm to ensure precise contact between the patient's bone and the fracture plate. Modelling of the patient's CT data were utilized to establish a threshold range of 200 to 350 HU, enabling the extraction of valid data corresponding to the affected bone region. Subsequently, a 3D reconstruction process was employed to convert the extracted data into a 3D model suitable for design applications. To address surface irregularities arising from CT scan slice segmentation during the 3D reconstruction process, the Smooth Factor function in Mimics software was applied at a fixed value of 0.6.

A Figure presenting all bones modelled as well as individual fixation plates should be added to the manuscript.

We appreciate your comment. We assume that figure 1A presents simulation model of the fractured tibia aligned correctly. In figure 1B, the figure presents fracture plate designing upon aligned simulation model. Figure 1C presents individual fixation plates. We noticed that the figure legend may not be descriptive enough and specified individual fixation plate.

While the authors claim “The optimal design was derived during this process by simulating the type and size of screws for fixation (…)”, there are no details regarding how optimal design was selected.

Thank you for your thorough review of this paper.

A simulation of the optimal screw insertion position was conducted utilizing three-dimensional morphology and medical imaging, ensuring the screw was positioned in a stable location and its length was accurately determined. The insertion pathway was meticulously planned to avoid critical structures such as nerves and blood vessels adjacent to the fracture site, with the screw insertion angle optimized accordingly. Through this optimization process, it was confirmed that the design prevents errors in the direction or depth of screw insertion when the plate is positioned on the bone, thereby enhancing fixation stability. This process is conducted as illustrated in the attached image below.

We agree that the detail lacks in description of the simulation of the optimal screw insertion and have revised the Participants section of the Materials and methods. Revised “Designing the Fracture Plate” of the materials and method section starts as below:

The simulation of the optimal screw insertion position was conducted by utilizing three-dimensional morphology and medical imaging. The screw insertion pathway was designed by simulating the type (cortical or locking screw) and appropriate length of screws for fixation, as well as determining the angle of screw fixation to be in accordance to AO principles of fracture management, such as achieving lag screw technique perpendicular to the fracture site as possible to the bone, ensuring avoidance of the adjacent neurovascular structure as well as stable fixation of the fracture plate within the body.

“known for its excellent precision among metal 3D printing products” – this description is redundant.

Thank you for your thorough review of this paper. We have deleted redundant part.

Results

It is unclear what authors mean by the “size” of the screw. Is it Screw diameter?

Thank you for your thorough review of this paper and pointing this out. The size of the screw means diameter. Therefore we have revised the “size” of the screw as its “diameter”.

General dimensions of the fixation plates should be presented in any of included Figures.

We appreciate your comment. However, due to the characteristics of the personalized 3D printed fixation plate, general dimensions of the fixation plates are hard to be generalized. However, the thickness of the fracture plate was designed to be 3.5mm to ensure stable support for the fixing screws in distal tibia area. This thickness is mentioned in the “Pure Titanium Fracture Plate section” of the Results.

Figures 3a and 3b presenting biomechanical testing setup should be presented in materials and methods section.

Thank you for your thorough review of this paper. We added reference to the figure 3a and 3b to the “Biomechanical Testing” section of the materials and methods.

“the fracture plate remained undamaged, deformed” – did the authors mean that plates were undamaged and undeformed over time?

Thank you for your thorough review of this paper. In “Clinical Efficacy Assessment” area, we meant that plates were undamaged and undeformed throughout 2 year follow up period.

Therefore, we revised the sentence to be clear as follows:

Over the follow up period, in nine out of 10 tibiae, the fracture plate remained undamaged, deformed, and was securely fixed within the body.

Reviewer #2:

First, the study's primary objective is not clearly presented. The authors claim in the Introduction that “To date, no reports have been documented on the use of 3D printed pure titanium plates specifically designed for the management of tibial fractures.” seems overly definitive. For instance, studies such as Duan et al. (Study on the efficacy of 3D printing technology combined with customized plates for the treatment of complex tibial plateau fractures. JOSR, 2024,) have investigated 3D-printed pure titanium plates for tibial fractures. The manuscript appears to narrow its focus to 3D printed locking screw but fails to explicitly state whether it aims to compare 3D-printed pure titanium plates with titanium alloy products or with traditionally manufactured locking mechanisms. Such comparisons are neither conducted nor addressed in the Discussion. For example, products like the DePuy Synthes 4.5/5.0 narrow locking compression plates are fabricated using pure titanium (although without 3D printing), and comparisons with related studies could have strengthened the discussion.

Thank you for your thorough review of this paper. We appreciate your comment. The pure titanium plates utilized in this study exhibit distinct characteristics that differentiate them from existing products. Specifically, the plates developed in this study are patient-specific products manufactured using additive manufacturing techniques with a metal 3D printer, rather than conventional machining methods. Unlike off-the-shelf products, these plates are designed based on the patient’s medical imaging data, allowing for precise conformation to the cortical bone’s outer contours and resulting in plates that perfectly fit the curvature of the patient’s bones.

When comparing the plates used in this study with Synthes products, two significant differences stand out. The first is the manufacturing method, and the second is the ability to produce patient-specific products. These differences highlight the key advantages of the plates developed in this study.

Furthermore, the plates produced in this study demonstrate technological distinctions even when compared with existing 3D-printed plates. Optimized process control techniques were employed during additive manufacturing to precisely adjust the parameters of the layering process, significantly enhancing the mechanical strength of the products. Notably, more than 10 process variables were meticulously controlled to overcome the mechanical limitations of pure titanium. Additionally, the process control took into account phase transformations observed in the chemical composition and microstructure, enabling fine-tuning of the mechanical properties to further increase strength.

This technological approach enabled the production of products with superior mechanical strength, suitable for clinical application. This study aims to present the potential of a novel therapeutic tool by overcoming the physical limitations of pure titanium and leveraging differentiated manufacturing methods and technical advantages.

We are aware of our limitation on no control group to compare, however we tried to focus on our manufacturing strength and have revised manuscript in general.

Additionally, the study's aim to evaluate whether locking mechanisms can be effectively implemented using 3D-printed pure titanium is inadequately addressed, as the locking threads in this study were CNC-machined (Methodology: “the locking thread was processed through computer numerical control (CNC) precision machining”). This process alters surface properties and may obscure potential internal defects inherent to 3D printing. The authors need to justify how this limitation does not compromise the study's significance.

We appreciate your comment. In this study, screws utilized were designed with a locking mechanism rather than the conventional non-locking type, necessitating highly precise thread machining. The manufacturing method using metal 3D printing is not well-suited for producing reverse-pyramid-shaped screw threads, due to the following reasons:

Manufacturing Constraints: Fabricating reverse-pyramid-shaped screw threads with a 3D printer requires support structures to prevent collapse during the process. However, the narrow spacing between the threads complicates support removal, and the nature of metal additive manufacturing may result in partial fusion of the support structures to the threads.

Lack of Precision: Due to these limitations, threads produced by 3D printing may exhibit reduced precision, which can compromise the core functionality of the locking mechanism when applied clinically. This may undermine the benefits of the locking mechanism, which is designed to firmly secure the plate to the bone, minimizing micromovements. A decrease in thread precision could negatively affect the friction and integration between the plate and bone, thereby impairing fixation stability.

Accordingly, this study employed CNC precision machining to fabricate accurate screw threads, thereby maximizing the effectiveness of the locking mechanism. This approach was essential to achieving reliable results consistent with the study's objectives.

Altogether, we have revised the introduction section to be more precise, and to point out our primary contribution in manufacturing of locking head thread and the clinical evaluation of the resulting implants.

Similar questions have been explored, as noted in MacLeod et al. (3D printed locking osteosynthesis screw threads have comparable strength to machined or hand‐tapped screw threads. JOR, 2020), which demonstrated that 3D-printed locking osteosynthesis screw threads have comparable strength to machined or hand-tapped screw threads. The current study should better clarify its novelty in light of this existing knowledge. Please clearly define the study objective, ensuring alignment with the experimental design and discussion.

Thank you for your thorough review of this paper. According to Dragos, (Risk factors of titanium locking plate osteosynthesis. In: 2015 E-Health and Bioengineering Conference (EHB). IEEE, 2015) the paper concluded, "Our results demonstrate that the locking plates made of commercially pure titanium, characterized by high hardness of the material, are susceptible to breakage if bone consolidation is delayed." It should be seen that as the purity of titanium increases, we should be careful of breakage. And with MacLeod et al. (3D printed locking osteosynthesis screw threads have comparable strength to machined or hand‐tapped screw threads. JOR, 2020), very novel and remarkable study conducted, but the thread geometries were not performed in head of the screw, distinct from locking head screw used in clinics.

Locking screws of the commercial products have mechanism of locking screw and plate with the screw head thread as below supplement figure.

Therefore, our study has significance in additive manufacturing of pure titanium locking threads for locking screw heads, and application to the clinical evaluation and outcome.

To me, the primary contribution of this manuscript appears to lie in the additive manufacturing of locking screws and the clinical evaluation of the resulting implants. However, the study's applicability is limited by the lack of detailed follow-up data to assess long-term clinical outcomes, and insufficient investigation into the fabrication process, particularly regarding post-processing techniques to address stiffness and micro-cracks inherent to 3D-printed metals. Validation of equivalence to traditionally manufactured screws in terms of fatigue resistance and mechanical reliability is missing.

We appreciate your comment. We agree that our study is relatively short term follow up period, and limitation to this study was added with a need for larger, multicenter studies with long term follow up data. We also agree to the point that insufficient description of materials and methods section, as well pointed out in other reviewer comments. More detailed information was provided in Materials and methods section in revised manuscript.

Additionally, our study has implications for improving mechanical properties of pure titanium by repeated heat treatment. (JUNG, Jae Hyun, et al. Improvement of mechanical properties of cast pure titanium by repeated heat treatment. Materials Science and Technology, 2019) This is added to the introduction section with reference added, and “fabrication of pure titanium fracture plate” section also describes two post-treatments performed to improve mechanical properties of fracture plate.

The manuscript lacks a robust discussion of the fixation plate design optimization. From Figure 4, it appears that the plate failure at the bending location could be mitigated by increasing the material thickness in this high-stress region. Engineers typically refine such designs using finite element analysis (FEA), which should be leveraged to optimize the plate's mechanical properties and maximize the advantages of additive manufacturing. While the study demonstrates the use of 3D printing for anatomical customization, it misses the opportunity to explore how this technology could enhance mechanical performance through material redistribution or gradient properties in high-risk areas.

We appreciate your comment. Our study focuses on evaluating the technical feasibility of patient-specific plates manufactured from pure titanium, with the primary objective of assessing their fundamental mechanical performance to ensure stable retention within the body. Consequently, rather than employing optimization design through finite element analysis (FEA), the plate was fabricated with uniform thickness to conform to the curvature of the bone. As such, FEA analysis was not within the scope of this study. The redistribution of material or incorporation of gradient properties through FEA to enhance mechanical performance in high-risk regions of the plate represents a highly significant proposal. We also think that this is a very crucial point. Future studies will actively explore this approach, as it is anticipated to contribute to the optimization of plate design and the comprehensive evaluation of the potential of additive manufacturing technologies.

The study conducted a static 4-point bending test and a dynamic 4-point fatigue test as per ASTM F382-17. However, ASTM F382-17 pertains to fatigue testing, not bending tests.

Thank you for your thorough review of this paper. ASTM F382-17 encompasses two types of tests. ASTM F382-17 A1 addresses the procedures for static four-point bending tests, while ASTM F382-17 A2 outlines the methods for dynamic four-point bending tests (fatigue testing). We have attached the relevant details attached below as a supplement material.

Al.l Scope:

Al.1.1 This test method describes methods for single cycle bend  testing  in  order  to  determine  the  intrinsic,  structural properties of metallic bone plates. The test method measures the bending stiffness, bending structural stiffness, and bending strength of bone plates.

A2.1. Scope

A2.1.1  This  test  method  describes  methods  for  bending fatigue testing in order to determine intrinsic, metallic bone plate structural properties. This test method may be used to determine  the  fatigue  life  at  a  specific  or  over  a  range  of maximum bending moment levels, or to estimate the fatigue strength for a specified number of fatigue cycles of a bone plate.

Reviewer 2 Report

Comments and Suggestions for Authors

Thank you for inviting me to review the manuscript titled Clinical Efficacy of 3D Printed Pure Titanium Fracture Plates with Locking Screw Systems in Distal Tibia Fractures. This manuscript explores a specialized topic in the application of 3D printing technology for orthopedic implants and presents some interesting information. However, there are significant concerns that require addressing to enhance the study's contribution to the field.

First, the study's primary objective is not clearly presented. The authors claim in the Introduction that “To date, no reports have been documented on the use of 3D printed pure titanium plates specifically designed for the management of tibial fractures.” seems overly definitive. For instance, studies such as Duan et al. (Study on the efficacy of 3D printing technology combined with customized plates for the treatment of complex tibial plateau fractures. JOSR, 2024,) have investigated 3D-printed pure titanium plates for tibial fractures. The manuscript appears to narrow its focus to 3D printed locking screw but fails to explicitly state whether it aims to compare 3D-printed pure titanium plates with titanium alloy products or with traditionally manufactured locking mechanisms. Such comparisons are neither conducted nor addressed in the Discussion. For example, products like the DePuy Synthes 4.5/5.0 narrow locking compression plates are fabricated using pure titanium (although without 3D printing), and comparisons with related studies could have strengthened the discussion. Additionally, the study's aim to evaluate whether locking mechanisms can be effectively implemented using 3D-printed pure titanium is inadequately addressed, as the locking threads in this study were CNC-machined (Methodology: “the locking thread was processed through computer numerical control (CNC) precision machining”). This process alters surface properties and may obscure potential internal defects inherent to 3D printing. The authors need to justify how this limitation does not compromise the study's significance.

Similar questions have been explored, as noted in MacLeod et al. (3D printed locking osteosynthesis screw threads have comparable strength to machined or hand‐tapped screw threads. JOR, 2020), which demonstrated that 3D-printed locking osteosynthesis screw threads have comparable strength to machined or hand-tapped screw threads. The current study should better clarify its novelty in light of this existing knowledge. Please clearly define the study objective, ensuring alignment with the experimental design and discussion.

To me, the primary contribution of this manuscript appears to lie in the additive manufacturing of locking screws and the clinical evaluation of the resulting implants. However, the study's applicability is limited by the lack of detailed follow-up data to assess long-term clinical outcomes, and insufficient investigation into the fabrication process, particularly regarding post-processing techniques to address stiffness and micro-cracks inherent to 3D-printed metals. Validation of equivalence to traditionally manufactured screws in terms of fatigue resistance and mechanical reliability is missing.

The manuscript lacks a robust discussion of the fixation plate design optimization. From Figure 4, it appears that the plate failure at the bending location could be mitigated by increasing the material thickness in this high-stress region. Engineers typically refine such designs using finite element analysis (FEA), which should be leveraged to optimize the plate's mechanical properties and maximize the advantages of additive manufacturing. While the study demonstrates the use of 3D printing for anatomical customization, it misses the opportunity to explore how this technology could enhance mechanical performance through material redistribution or gradient properties in high-risk areas.

The study conducted a static 4-point bending test and a dynamic 4-point fatigue test as per ASTM F382-17. However, ASTM F382-17 pertains to fatigue testing, not bending tests.

Author Response

(The authors gave the same response as above.)

Round 2

Reviewer 1 Report

Comments and Suggestions for Authors

The authors have referred to all of the comments presented in the previous review process. This allowed to appropriately increase the quality of the paper on the level allowing it to be recommended to be published in Medicina. There are only some minor adjustments required to be handled by the editorial board (the lack of spaces between value and unit etc.).

Best wishes to authors in their further research.

Author Response

Response to the Reviewers

Dear Reviewers,

We deeply appreciate your comments and questions, and believe that they have allowed us to significantly improve our revised manuscript. 

The reviewer’s comments or questions were reiterated in italics, and we have addressed them point-by-point. All changes in the revised manuscript are shown using highlights. 

Thank you again for your consideration.

Sincerely yours

Reviewers Comments:

Reviewer #1: 

The authors have referred to all of the comments presented in the previous review process. This allowed to appropriately increase the quality of the paper on the level allowing it to be recommended to be published in Medicina. There are only some minor adjustments required to be handled by the editorial board (the lack of spaces between value and unit etc.).

Best wishes to authors in their further research.

  • Thank you for thoughtful review and comments.

Reviewer #2: 

While it is well established in the literature that 3D-printed patient-specific implants differ from conventional CNC or casting-based products, this does not appear to be the main contribution of the current study. Additive manufacturing is simply one of several digital manufacturing methods capable of producing patient-specific implants. To me, the novelty of this work lies in the use of 3D-printed pure titanium combined with CNC-manufactured locking screws and the subsequent clinical verification of these implants, rather than the material, process, or design side of the technology. Unfortunately, the study design deviates from adequately highlighting or validating these contributions. Instead, it relies on general fixation plate mechanical testing and a limited clinical follow-up cohort.

The authors could provide more sufficient study design to support its novelty, such as the superiority of locking screw mechanisms, programmable mechanical strength, or generalizability (diverse patient populations).

  • Locking Screw Mechanism and Novelty of the Study

The superiority of the locking screw mechanism has been well established in numerous previous studies, as supported by the cited references [12-14]. However, traditional 3D printing techniques have failed to implement such locking mechanisms due to the brittleness of titanium alloys. Moreover, plates made of pure titanium have historically faced limitations in their application for fracture fixation due to their insufficient mechanical strength.

In this study, we addressed these challenges by successfully utilizing 3D-printed pure titanium to precisely implement screw threads while enhancing the mechanical strength of the material. This advancement enabled the clinical application of the plates, demonstrating the novelty of this approach. Importantly, the method developed in this study is versatile and can be applied to plates of various shapes, allowing for generalization across different clinical scenarios.

Based on this perspective, we have added the following text to the Discussion section:

"This study demonstrates a novel approach by overcoming the limitations of titanium alloys in 3D printing and the mechanical weaknesses of pure titanium plates. By implementing precise screw threads and strengthening the material through optimized manufacturing processes, we have achieved sufficient mechanical properties for clinical application. Furthermore, the versatility of this approach enables its application to various plate geometries, underscoring its potential for broader clinical utility. "

The authors also emphasized the importance of "optimized process control techniques" in the additive manufacturing process, claiming that these controls significantly enhanced the mechanical strength of the implants. They highlight the meticulous adjustment of over 10 process variables, including considerations of phase transformations and microstructure refinement. However, the manuscript does not present any experimental evidence or detailed analysis demonstrating how these optimizations improve energy beam-metal powder interactions, enhance lattice homogeneity, or ultimately strengthen the material and locking screws. Also, the statement "layering process significantly enhances the mechanical strength of the products" is not adequately substantiated by the data presented or current literature.

  • We deeply agree with the reviewer’s concern regarding the lack of evidence supporting the effects of "optimized process control techniques" and the insufficient substantiation of the claim that "the layering process enhances the mechanical strength of the products." To address these issues, we have incorporated additional data and provided more detailed explanations as follows:

Background and Technological Basis of the Study

The bone plates used in this study were developed using a metal microstructure control additive manufacturing method, a technology transferred from the Korea Institute of Industrial Technology (KITECH). The core of this technology lies in controlling laser scan speed, spot distance, and exposure time to produce microstructures with the desired properties. Through extensive testing and optimization, we derived the conditions most suitable for the product.

Optimization of Laser Scanning Conditions

The study utilized optimized parameters, including scan speeds of 250–500 mm/s, spot distances of 40–100 μm, and exposure times of 80–400 μs, to stably induce isotropic α-phase microstructures in pure titanium. The optimal conditions are visually illustrated in [Figure 1] and [Figure 2].

Layering Process and Mechanical Strength

We controlled the layer thickness (20–100 μm) and average particle size (10–100 μm) to manipulate the grain size and orientation of the microstructures. While hardness decreased slightly, the material exhibited stable mechanical properties in all directions, ensuring its suitability for clinical applications.

Energy Density and Microstructure Homogeneity

By optimizing energy density, we minimized uneven microstructures, avoided peripheral melting, and reduced residual stress. This approach achieved a balance between strength and ductility, which is critical for clinical performance.

Experimental Validation and Additional Data

Experiments were designed based on patented protocols. As demonstrated in [Figure 3] and [Figure 4], EBSD (Electron Backscatter Diffraction) and Vickers hardness data quantitatively analyzed the relationship between microstructure and mechanical properties. These findings substantiate the enhanced properties resulting from the optimized manufacturing process.

The process of manufacturing pure titanium at KITECH has been reported in the following two studies:

  1. Jung, J.H.; Choi, J.; Lee, B.-S.; Na, T.-W.; Kim, H.G.; Park, H.-K. Improvement of mechanical properties of cast pure titanium by repeated heat treatment. Materials Science and Technology, 2019, 35(3), 248–252.
  2. Na, T.-W.; Kim, W.R.; Yang, S.-M.; Kwon, O.; Park, J.M.; Kim, G.-H.; et al. Effect of laser power on oxygen and nitrogen concentration of commercially pure titanium manufactured by selective laser melting. Materials Characterization, 2018, 143, 110–117.

The following content has been added to the Methods section:

"The bone plates were manufactured using a microstructure-controlled additive manufacturing method licensed from the Korea Institute of Industrial Technology. This technology optimizes laser scan speed (250–500 mm/s), spot distance (40–100 μm), and exposure time (80–400 μs) to induce isotropic α-phase microstructures in pure titanium. Additionally, layer thickness (20–100 μm) and particle size (10–100 μm) were controlled to improve grain size and orientation. Energy density adjustments were made to enhance microstructure homogeneity and reduce residual stress. Experimental validation using electron backscatter diffraction and vickers hardness testing confirmed the relationship between the optimized parameters and the mechanical properties of the material. [12,13]"

  1. Jung, J.H.; Choi, J.; Lee, B.-S.; Na, T.-W.; Kim, H.G.; Park, H.-K. Improvement of mechanical properties of cast pure titanium by repeated heat treatment. Materials Science and Technology, 2019, 35(3), 248–252.
  2. Na, T.-W.; Kim, W.R.; Yang, S.-M.; Kwon, O.; Park, J.M.; Kim, G.-H.; et al. Effect of laser power on oxygen and nitrogen concentration of commercially pure titanium manufactured by selective laser melting. Materials Characterization, 2018, 143, 110–117.

Reviewer 2 Report

Comments and Suggestions for Authors

While it is well established in the literature that 3D-printed patient-specific implants differ from conventional CNC or casting-based products, this does not appear to be the main contribution of the current study. Additive manufacturing is simply one of several digital manufacturing methods capable of producing patient-specific implants. To me, the novelty of this work lies in the use of 3D-printed pure titanium combined with CNC-manufactured locking screws and the subsequent clinical verification of these implants, rather than the material, process, or design side of the technology. Unfortunately, the study design deviates from adequately highlighting or validating these contributions. Instead, it relies on general fixation plate mechanical testing and a limited clinical follow-up cohort.

The authors could provide more sufficient study design to support its novelty, such as the superiority of locking screw mechanisms, programmable mechanical strength, or generalizability (diverse patient populations).

The authors also emphasized the importance of "optimized process control techniques" in the additive manufacturing process, claiming that these controls significantly enhanced the mechanical strength of the implants. They highlight the meticulous adjustment of over 10 process variables, including considerations of phase transformations and microstructure refinement. However, the manuscript does not present any experimental evidence or detailed analysis demonstrating how these optimizations improve energy beam-metal powder interactions, enhance lattice homogeneity, or ultimately strengthen the material and locking screws. Also, the statement "layering process significantly enhances the mechanical strength of the products" is not adequately substantiated by the data presented or current literature.

Author Response

(The authors gave the same response as above.)
